# A Porcine DNMT1 Variant: Molecular Cloning and Generation of Specific Polyclonal Antibody

**DOI:** 10.3390/genes14071324

**Published:** 2023-06-23

**Authors:** Lin Zhu, Jiayun Wang, Yanbing Zhang, Xiao Xiang, Ke Liu, Jianchao Wei, Zongjie Li, Donghua Shao, Beibei Li, Zhiyong Ma, Yafeng Qiu

**Affiliations:** Shanghai Veterinary Research Institute, Chinese Academy of Agricultural Sciences, 518 Ziyue Road, Shanghai 200241, China; taishanzl966@163.com (L.Z.); wangjiayun200@163.com (J.W.); zhangyanbing@shzu.edu.cn (Y.Z.); xiangxiao119@126.com (X.X.); liuke@shvri.ac.cn (K.L.); jianchaowei@shvri.ac.cn (J.W.); lizongjie@shvri.ac.cn (Z.L.); shaodonghua@shvri.ac.cn (D.S.); lbb@shvri.ac.cn (B.L.); zhiyongma@shvri.ac.cn (Z.M.)

**Keywords:** porcine alveolar macrophages, DNMT1, synthetic peptide, polyclonal antibody

## Abstract

DNA methyltransferase 1 (DNMT1), the first-identified DNA methyltransferase in mammals, has been well studied in the control of embryo development and somatic homeostasis in mice and humans. Accumulating reports have demonstrated that DNMT1 plays an important role in the regulation of differentiation and the activation of immune cells. However, little is known about the effects of porcine DNMT1 on such functional regulation, especially the regulation of the biological functions of immune cells. In this study, we report the cloning of *DNMT1* (4833 bp in length) from porcine alveolar macrophages (PAMs). According to the sequence of the cloned DNMT1 gene, the deduced protein sequence contains a total of 1611 amino acids with a 2 amino acid insertion, a 1 amino acid deletion, and 12 single amino acid mutations in comparison to the reported DNMT1 protein. A polyclonal antibody based on a synthetic peptide was generated to study the expression of the porcine DNMT1. The polyclonal antibody only recognized the cloned porcine DNMT1 and not the previously reported protein due to a single amino acid difference in the antigenic peptide region. However, the polyclonal antibody recognized the endogenous DNMT1 in several porcine cells (PAM, PK15, ST, and PIEC) and the cells of other species (HEK-293T, Marc-145, MDBK, and MDCK cells). Moreover, our results demonstrated that all the detected tissues of piglet express DNMT1, which is the same as that in porcine alveolar macrophages. In summary, we have identified a porcine DNMT1 variant with sequence and expression analyses.

## 1. Introduction

DNA methylation is a major epigenetic modification of the genome that plays a vital role in the regulation of gene expression [1,2,3]. In mammals, there are three major DNA methyltransferases (DNMTs), which catalyze the covalent binding of a methyl group provided by S-adenosylmethionine to cytosine in DNA [4,5,6]. DNMT1, the first-identified DNA methyltransferase, is known to maintain DNA methylation patterns [7,8]. In contrast, DNMT3A and DNMT3B are de novo DNMTs, responsible for establishing methylation patterns [2]. Of these three DNMTs, DNMT1 is the most abundant methyltransferase in mammalian somatic cells [9,10]. DNMT1 binds to the replication fork during DNA replication and has a high affinity for semi-methylated DNA double strands, which allows the enzyme to copy the methylation pattern after DNA replication [11,12,13]. Accumulating evidence has implied that aberrant expression of DNMT1 causes alteration of the DNA methylation pattern, which is linked to the occurrence and development of several cancers [14,15,16,17]. Therefore, DNMT1 has been identified as a therapeutic target for the treatment of some cancers [18,19]. 

Because DNA methylation also regulates the differentiation and functions of immune cells [20], changes in DNA methylation patterns dysregulate these functions, causing immune system diseases. Therefore, controlling DNA methylation may offer an avenue for treating immune-related diseases [21]. Macrophages are important components of the immune system, facilitating the maintenance of tissue homeostasis and preventing microbial infections [22]. However, the phenotypes of macrophages can be altered by different stimuli to become classically activated (M1) or alternatively activated (M2) macrophages [23], leading to various diseases [24]. For example, M1-polarized macrophages promote the development of atherosclerosis. Although the underlying mechanism by which M1 macrophages are polarized requires further study, DNMT1 plays a key role in the regulation of M1 macrophage polarization by alteration of inflammation [25]. 

Epigenetic regulation by DNMT1 has been well studied in humans and mice, but the function of porcine DNMT1 is less well understood. We are particularly interested in the role of epigenetic regulation by DNA methylation in the immune response of porcine macrophages. In a previous study, we identified DNMT3B2 as the predominant isoform of DNMT3B in porcine alveolar macrophages and demonstrated its potential role in the regulation of lipopolysaccharide (LPS)-stimulated tumor necrosis factor α (TNF-α) expression [26]. In this study, we characterized DNMT1 in porcine alveolar macrophages. Surprisingly, we identified a porcine DNMT1 variant with 1611 amino acids with sequencing and protein analyses.

## 2. Materials and Methods

### 2.1. Cells and Tissues

Pudong White piglets were purchased from the Shanghai Academy of Agricultural Sciences (Shanghai, China) at around 30 days of age, euthanized, and dissected to obtain porcine alveolar macrophages (PAMs) and various tissue samples (heart, liver, spleen, lung, kidney, small intestine, thymus, inguinal lymph nodes, submaxillary lymph nodes, mesenteric lymph nodes, tonsils, and hilar lymph nodes). The PAMs were isolated, as reported in our previous study [27]. Briefly, the pig lungs were washed with 400 mL of PBS containing 1 mM EDTA. The recovered cell suspension was spun at 500× *g* for 10 min, and the cell precipitate was resuspended in RPMI 1640 containing 10% fetal bovine serum (FBS), penicillin, streptomycin, and GlutaMAX™ Supplement (all purchased from Thermo Fisher Scientific, Shanghai, China). Furthermore, the tissue samples were freshly generated by using around 100 mg of each tissue (heart, liver, spleen, lung, kidney, small intestine, thymus, inguinal lymph nodes, submaxillary lymph nodes, mesenteric lymph nodes, tonsils, and hilar lymph nodes). Subsequently, the aliquoted tissues were homogenized and lysed for later Western blotting analysis. All the experiments were performed in accordance with procedures approved by the Animal Care and Use Committee of the Shanghai Veterinary Research Institute (IACUC No: Shvri-po-2016060501), Chinese Academy of Agriculture Science. Human embryonic kidney (HEK-293T) cells, Marc-145 cells, mouse neuroblastoma (N2a) cells, baby hamster kidney (BHK-21) cells, Madin–Darby bovine kidney (MDBK) cells, Madin–Darby canine kidney (MDCK) cells, porcine kidney (PK)-15 cells, swine testicular (ST) cells, and porcine iliac artery endothelial cells (PIECs) were maintained in Dulbecco’s Modified Eagle Medium (Thermo Fisher Scientific) supplemented with 10% FBS (Thermo Fisher Scientific) at 37 °C in a 5% CO_2_ atmosphere.

### 2.2. Cloning of Porcine DNMT1 and Sequence Analysis

The total RNA was extracted from the PAMs (at least 1.0 × 10^6^ cells) with TRIzol™ Reagent (Accurate Biotechnology, Changsha, China) [28], and the cDNA was prepared with Super Script II Reverse Transcriptase (Thermo Fisher Scientific). Five pairs of primers for cloning the porcine DNMT1 gene (shown in Appendix A) were designed on the basis of the porcine *DNMT1* cDNA sequence reported in the GenBank database (NM_001032355.1). All the PCR products were sequenced with gene-specific primers. All the images of agarose gel electrophoresis were captured with the Image Lab Software version 5.1 (Bio-Rad Laboratories, Hercules, CA, USA). 

The amino acid sequences of the human (NP_001124295.1), mouse (NP_001300940.1), porcine (NP_001027526.1), and cloned porcine DNMT1 proteins were aligned with Clustal V and edited with Genedoc. A phylogenetic tree was constructed from the available DNMT1 proteins with the neighbor-joining method in MEGA version 6.06 [29].

### 2.3. Generation of Polyclonal Anti-Porcine DNMT1 Antibody

A polyclonal antibody directed against porcine DNMT1 was generated as described in a previous study [30]. Briefly, antigenic peptides from the cloned porcine DNMT1 were predicted by using an online tool (https://novoprolabs.com/tools/peptide-antigen-design, accessed on 30 June 2020). According to the predicted structure of the cloned porcine DNMT1 (https://swissmodel.expasy.org/interactive, accessed on 30 June 2020), a peptide of 15 amino acids (SSPVKRPRKEPVDED) exposed outside was selected. The peptide was then synthesized chemically and conjugated with keyhole limpet hemocyanin (KLH) as the carrier protein. Two large New Zealand rabbits were immunized five times with the peptide–KLH conjugate combined with complete or incomplete Freund’s adjuvants. The rabbit experiments were approved by the Institutional Animal Care and Use Committee of Shanghai Veterinary Research Institute (IACUC No.: Shvri-po-201606 0501) and conformed to the Guidelines for the Humane Treatment of Laboratory Animals (Ministry of Science and Technology of the People’s Republic of China, Policy No. 2006398).

### 2.4. Plasmid Transfection

The cloned porcine *DNMT1* cDNA was inserted into the p3×Flag-CMV-14 vector (Sigma, St. Louis, MO, USA) and designated pFlag-DNMT1 (cloned). In the meantime, the reported porcine *DNMT1* cDNA (NM_001032355.1) was synthesized and cloned into the p3×Flag-CMV-14 vector and the named pFlag-DNMT1 (NM_001032355.1). To determine the specificity of the anti-porcine DNMT1 antibody, HEK-293T cells were grown to 70–80% confluence and transfected with the pFlag–DNMT1 (cloned) or pFlag vector plasmid using Lipofectamine™ 2000 Transfection Reagent (Thermo Fisher Scientific) according to the manufacturer’s instructions. To identify the porcine DNMT1 sequence recognized by the polyclonal anti-porcine DNMT1 antibody, BHK-21 cells were grown to 70–80% confluence and transfected with the plasmids pFlag–DNMT1 (cloned), pFlag–DNMT1 (NM_001032355.1), or pFlag–vector, as described above. After transfection for 24 h, the samples were collected for further analysis.

### 2.5. Western Blotting

Briefly, membranes containing the transferred protein samples were blocked with 5% skim milk for 2 h at room temperature. They were then incubated overnight at 4 °C with the individual primary antibodies: anti-Flag (M2, Sigma; diluted 1:2000), anti-porcine DNMT1 (generated in this study; diluted 1:2000), or anti-β-actin (Sigma; diluted 1:10,000). The membranes were then incubated for 1 h at room temperature with the appropriate secondary antibody: horseradish peroxidase (HRP)-conjugated goat anti-mouse IgG (Abcam; diluted 1:5000) or HRP-conjugated goat anti-rabbit IgG (Abcam; diluted 1:10,000). The membrane was then treated with HRP substrate according to the instructions for Enhanced Chemiluminescence (ECL) Reagent (Pierce) and exposed to X-ray film. Images were captured with the Gel Doc™ EZ System (Bio-Rad Laboratories, USA).

### 2.6. Polymerase Chain Reaction (PCR)

The total DNA from the PAM, PK15, ST, and PIEC cells was extracted with a cell DNA Isolation Mini Kit (Vazyme, Nanjing, China). DNA fragments containing antigenic peptides and inserted or deleted regions were amplified by PCR using the extracted DNA samples above. All PCR products were sequenced with gene-specific primers. All the images of agarose gel electrophoresis were captured with the Image Lab Software version 5.1 (Bio-Rad Laboratories, Hercules, CA, USA). The specific primers are shown in Appendix A. 

## 3. Results

### 3.1. Molecular Cloning of Porcine DNMT1 cDNA and Multiple Sequence Alignment of DNMT1 from Different Species

To characterize the expression profile of DNMT1 in porcine alveolar macrophages, we first cloned the *DNMT1* cDNA with reverse transcription (RT)-PCR on the basis of the reported porcine DNMT1 sequence (NM_001032355.1) in the GenBank database. Five overlapping gene fragments covering the *DNMT1* cDNA were obtained and sequenced (Appendix A). A 4833 bp full-length *DNMT1* cDNA was identified, and the deduced protein sequence contained 1611 amino acids. In comparison, the reported porcine *DNMT1* cDNA (NM_001032355.1) is 4830 bp long and encodes 1610 amino acids. Notably, the cloned *DNMT1* cDNA contained a 6 bp nucleotide insertion at 3980–3981 bp and a 3 bp nucleotide deletion at 4812–4814 bp relative to the GenBank-reported porcine *DNMT1* cDNA. We then assembled a multiple-sequence alignment of pig, human, and mouse DNMT1 proteins using full-length amino acid sequences. The cloned porcine DNMT1 was 99% identical to the previously reported porcine DNMT1 (NP_001027526.1), with a 2 amino acid insertion, a 1 amino acid deletion, and 12 single amino acid mutations (Figure 1). Furthermore, we compared the cloned porcine DNMT1 with the other eight porcine DNMT1s deposited in GenBank (Appendix A), which are predicted from the genomic sequence (NC_010444.4). Our results showed that the eight predicted porcine DNMT1s all had the same 2 insertion amino acids and 12 mutant amino acids as the cloned DNMT1; in comparison, the deleted region only existed in the 3 predicted porcine DNMT1 (XP_020937688.1, XP_020937695.1, and XP_020937699.1). Since the eight porcine DNMT1s remain predicted, we chose the cloned DNMT1, the reported porcine DNMT1 (NP_001027526.1), the human DNMT1 (NP_001124295.1), and the mouse DNMT1 (NP_001300940.1) for multiple sequence alignment. Moreover, our results showed that the cloned porcine DNMT1 was 88% and 75.3% identical to that of humans and of mice, respectively (Figure 1). Therefore, although the cloned porcine DNMT1 was highly homologous with the previously reported porcine DNMT1 (NP_001027526.1), the sequence length and the indicated insertion, deletion, and mutation amino acids suggested that the cloned DNMT1 differed from the reported porcine DNMT1.

### 3.2. Phylogenetic Analysis of DNMT1 Protein Sequences

To understand the relationships between the cloned porcine DNMT1 and those of other species, we performed a phylogenetic analysis of 20 deduced amino acid sequences from different species. Three clusters were identified: mammal, bird, and fish DNMT1. The cloned porcine DNMT1 clustered in the mammal group together with the DNMT1s of goats, sheep, cattle, horses, rabbits, humans, and monkeys but not with those of rodents. Therefore, consistent with the results of the multiple-sequence alignment, the two porcine DNMT1s were closer to human DNMT1 than to mouse DNMT1 (Figure 2).

### 3.3. Generation of Polyclonal Antibody against Porcine DNMT1 and Cross-Reactivity of the Antibody against DNMT1 from Different Species

To determine the expression profile of DNMT1 in PAMs, we next developed a polyclonal antibody using a synthetic antigen peptide (SSPVKRPRKEPVDED), predicted and selected on the basis of the cloned porcine DNMT1 protein. The rabbits were immunized with this synthetic peptide, and antisera was prepared after seven immunizations. We used affinity chromatography to further purify the polyclonal antibody from the collected antisera. The enzyme-linked immunosorbent assay (ELISA) titer of the purified antibody was >10^5^. We used Western blotting to examine its reaction with the cloned porcine DNMT1 ectopically expressed in HEK-293T cells. Consistent with the results obtained with an anti-Flag antibody, the polyclonal antibody specifically detected the expression of the cloned porcine DNMT1 (Figure 3A) and specifically detected a band similar to the ectopically expressed porcine DNMT1 consistent with endogenous human DNMT1 in the vector-transfected group (Figure 3A).

To detect the cross-reactivity of the antibody against DNMT1 from different species, we tested cell samples from humans, monkeys, pigs, mice, hamsters, cattle, and dogs with Western blotting. The polyclonal antibody recognized the endogenous DNMT1 in HEK-293T, Marc-145, PAM, MDBK, and MDCK cells, but not in N2a or BHK-21 cells (Figure 3B). To understand the different cross-reactivity of the antibody against DNMT1 proteins from various species, we further examined the peptide sequences of all the species described above. As shown in Figure 3C, the cloned porcine DNMT1, human DNMT1, monkey DNMT1, and dog DNMT1 were 100% identical to the synthetic peptide. However, one amino acid differed in the synthetic peptide relative to the GenBank porcine DNMT1 (R6G) and bovine DNMT1 (D15A). Notably, the synthetic peptide differed from mouse DNMT1 and hamster DNMT1 in five amino acids, which resulted in its lack of cross-reaction with mouse and hamster DNMT1 (Figure 3B). Overall, our data indicate that a polyclonal antibody was successfully generated to detect the expression of cloned porcine DNMT1 and DNMT1 in several other species.

### 3.4. Identification of Two Different Porcine DNMT1s with a Single Amino Acid Difference in the Antigenic Peptide Region

Although the polyclonal antibody cross-reacted with bovine DNMT1, with which it shared one amino acid difference (D15A), it was unclear whether the antibody could be used to detect the reported porcine DNMT1 (R6G). Therefore, we examined the reactivity of the antibody against the reported porcine DNMT1 (NM_001032355.1) with Western blotting. To exclude the endogenous DNMT1, the BHK-21 cells were transiently transfected with plasmid pFlag–vector, pFlag–DNMT1 (NM_001032355.1), or pFlag–DNMT1 (cloned). Surprisingly, the polyclonal antibody only reacted with the cloned porcine DNMT1 and not with the reported porcine DNMT1 (NM_001032355.1) (Figure 4A). These data indicate that the one-amino-acid difference (R6G) in the antigenic peptide region of the reported DNMT1 abolished its reactivity to our polyclonal antibody.

### 3.5. Identification of a Porcine DNMT1 Variant in Different Porcine Cells and Tissues

As shown in the results described above, our polyclonal antibody recognized the cloned porcine DNMT1 but not the previously reported protein. This result motivated us to determine the expression profile of DNMT1 in other porcine cells (PK15, ST, and PIEC) with Western blotting. Bands of the same size were detected in other porcine cells as were detected in PAMs (Figure 4B), suggesting that the DNMT1 in those cells contained the same antigenic peptide. To confirm this result, we determined the sequence of the antigenic peptides in different porcine cells. Because the DNA sequence encoding the antigenic peptide is located in the same exon in all *DNMT1* genes, we used PCR to amplify the sequence of interest from the DNA of different porcine cells. Sequence analysis showed that all the amplified sequences were 100% identical to the cloned DNMT1. Furthermore, the nucleotide sequences and deduced amino acids of the peptide region in those cells were 100% identical to those of the synthetic antigen peptide (Figure 4C). We also examined the sequences of the inserted and deleted regions in the DNMT1 in those porcine cells. Consistent with our previous findings, our results showed that the DNMT1 in all the porcine cells analyzed had a 2 amino acid insertion and a 1 amino acid deletion relative to the GenBank-reported porcine DNMT1 (Figure 4D). Thus, the sequence and expression analyses revealed the presence of a porcine DNMT1 variant in these porcine cells that differed from the reported porcine DNMT1. Moreover, the expression profile of DNMT1 in different tissues of piglet was investigated by Western blotting. Our results showed that all the detected tissues express DNMT1, consistent with that in porcine alveolar macrophages (Figure 4E). Collectively, our results identify a different DNMT1 variant, relative to reported DNMT1 (NP_001027526.1), in various porcine cells and tissues.

## 4. Discussion

In this study, we identified a porcine DNMT1 variant containing 1611 amino acids, with a 2 amino acid insertion, a 1 amino acid deletion, and 12 single amino acid mutations relative to the previously reported porcine DNMT1 protein. A rabbit polyclonal antibody was generated using a 15-amino acid antigenic peptide, which contained an amino acid mutation relative to the reported porcine DNMT1 (NP_001027526.1). Our data demonstrate that the polyclonal antibody specifically recognized the cloned porcine DNMT1 but not the originally reported porcine DNMT1, which contained a single amino acid difference in the antigenic peptide region. Our data also show that this antibody recognized endogenous DNMT1 in other porcine cells and tissues, which contained the same amino acids as the synthetic peptide. This antibody also cross-reacted with the DNMT1 proteins of other species including humans, monkeys, dogs, and cattle. Collectively, these data confirm the identification of a DNMT1 variant from PAMs.

Because DNMT1 plays a crucial role in the regulation of the inflammatory response and macrophage polarization [25], it is important to understand the characteristics of DNMT1 in PAMs. Therefore, we cloned the *DNMT1* cDNA from PAMs. Unexpectedly, we detected a DNMT1 variant with 1611 amino acids, which has a 2 amino acid insertion, a 1 amino acid deletion, and 12 single amino acid mutations relative to the previously reported porcine DNMT1. In further analysis, we demonstrated the existence of the same insertion, deletion, and mutation amino acids in the other three deposited porcine DNMT1 (XP_020937688.1, XP_020937695.1, and XP_020937699.1), which are the predicted sequences from the genomic sequence (NC_010444.4). Although those predicted sequences need to be verified, this datum is important to our results. 

Fortunately, we have generated a polyclonal antibody to identify two different porcine DNMT1s with a single amino acid difference in the antigenic peptide region. Western blotting assay has confirmed that PK15, ST, and PIEC cell lines all express the DNMT1 variant. We also demonstrated that the DNA sequences encoding the antigenic peptide and the inserted and deleted regions of DNMT1 in all the porcine cell lines were 100% identical to the sequences in the DNMT1 cloned from the PAMs. Therefore, this porcine DNMT1 variant also exists in all the porcine cell lines tested. Notably, we have demonstrated that the various tissues of piglets are also robust in expressing the DNMT1 variant. Thus, this antibody has become an important tool to detect porcine DNMT1. 

In the meantime, this antibody is useful in detecting DNMT1 from other species including humans, monkeys, dogs, and cattle. However, it is not capable of recognizing the DNMT1 of rodents such as mice and hamsters. According to sequence analysis, it is understandable that either mouse DNMT1 or hamster DNMT1 contains five amino acid differences in the antigenic peptide. Interestingly, either mouse or hamster DNMT1 does not contain the sixth amino acid difference in antigenic peptide, but they do both contain the fifteenth amino acid difference. According to our results, the sixth amino acid is a key amino acid for determining the reactivity of this polyclonal antibody; in comparison, the fifteenth amino acid is not. Thus far, it is clear that the sixth amino acid is a key amino acid for deciding the reactivity of this polyclonal antibody. Whether or not there is another key amino acid needs further experimental analysis. 

The porcine DNMT1 variant was identified in PAMs from Pudong White pigs, which historically originate in Shanghai, China [31]. The indigenous Pudong White pig, which has the advantages of feed and production, has been identified as a distinctive genetic resource, with a unique genetic structure, and has been added to China’s livestock and poultry genetic resources listed for conservation by the Ministry of Agriculture of China [32]. In a phylogenetic analysis, Pudong White pigs clustered together with other Chinese pigs rather than with European pigs or the out-group [31]. Therefore, to know the expression profile of this DNMT1 variant in other Chinese pigs, European pigs, and the out-group, further studies are needed.

In somatic cells, canonical full-length DNMT1 is highly conserved in the same species. Some mutation(s) in canonical DNMT1 may alter its functions, causing DNMT1-related disorders. Hereditary sensory and autonomic neuropathy 1E (HSAN1E) has been shown to be associated with mutations in DNMT1 [33], and the naturally occurring S878F DNMT1 mutation is reported to result in elevated fetal hemoglobin levels [34]. An investigation of the single nucleotide polymorphisms (SNPs) of DNMT1 showed that mutations of DNMT1 are related to a variety of diseases, including cancers, heart disease, and autoimmune diseases [35,36,37,38,39,40]. Although there has been no report of DNMT1-related disorders in pigs, it is important to determine whether this variant contributes to any disease susceptibility or resistance in Pudong White pigs. It would also be interesting to know whether this variant contributes to the specific traits of feed and production in Pudong White pigs.

In this study, we have demonstrated a DNMT1 variant in the PAMs of Pudong White pigs. To the best of our knowledge, this is the first time that a DNMT1 variant has been identified in porcine somatic cells. How this variant functionally regulates the homeostasis of porcine somatic cells, such as macrophages, is still unclear, and future studies are necessary to address this issue.

## Figures and Tables

**Figure 1 genes-14-01324-f001:**
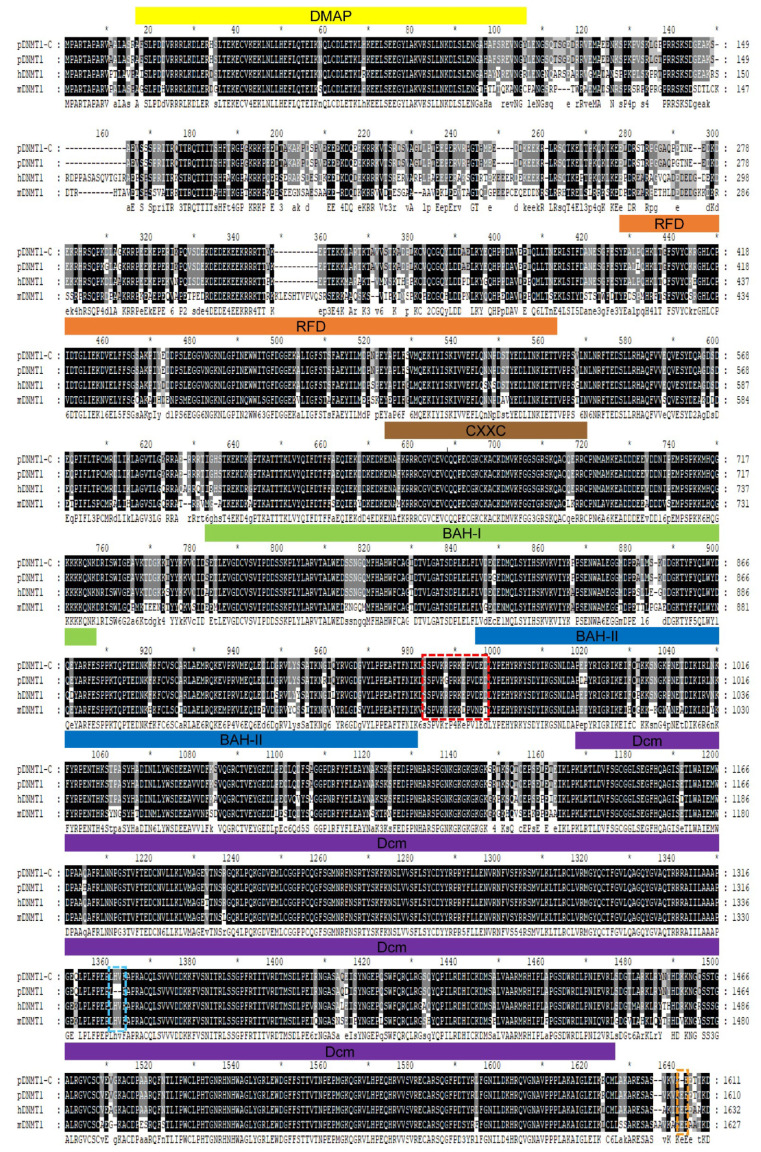
Multiple-sequence alignments of DNMT1 from different species. Sequences of pig (NP_001027526.1), human (NP_001124295.1), mouse (NP_001300940.1), and the cloned porcine DNMT1 proteins were aligned to determine the levels of homology. Black and gray shading highlights the sequence consistency and similarity across all selected sequences, respectively. The DMAP binding (yellow line), RFD (orange line), -CXXC-type zinc finger (brown line), BAH-1 (green line), BAH-2 (blue line), and Dcm (purple line) domains relative to the porcine DNMT1 (NP_001027526.1) are labeled. The red dashed box indicates the DNMT1 antigenic peptide, which we used for anti-DNMT1 antibody development. Blue and orange dashed boxes indicate inserted and deleted regions relative to the reported porcine DNMT1 (NP_001027526.1). To note that each interval of 10 amino acids is indicated as an asterisk.

**Figure 2 genes-14-01324-f002:**
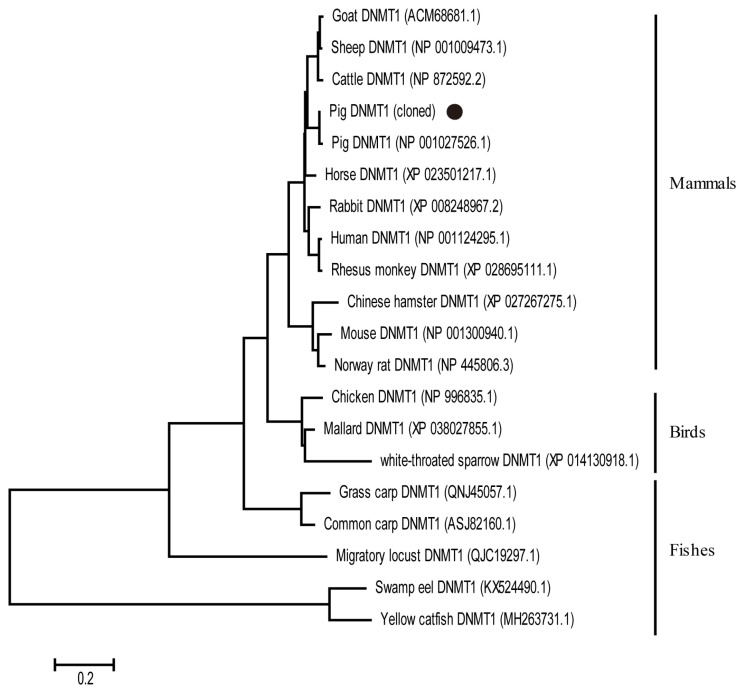
Phylogenetic analysis of DNMT1 proteins from various species. The phylogenetic tree was constructed from available DNMT1 proteins with the neighbor-joining method in MEGA version 6.06. Scale bar indicates the genetic distance. The cloned DNMT1 is marked with black dots.

**Figure 3 genes-14-01324-f003:**
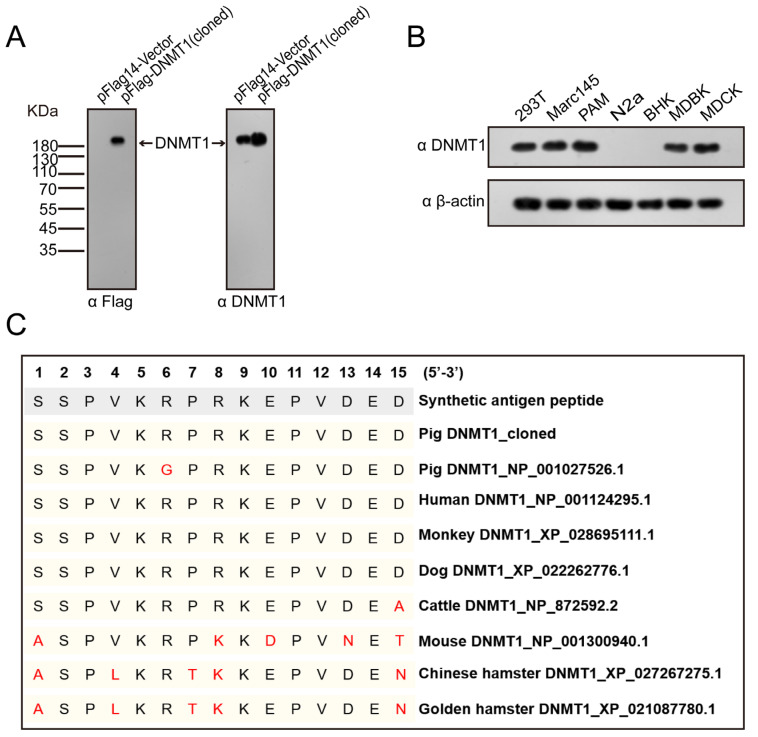
Cross-reactivity of the antibody against DNMT1 in various animal cells. (**A**) HEK-293T cells were transiently transfected with pFlag14–Vector or pFlag14–DNMT1 (cloned) plasmid. Cell lysates were analyzed with anti-Flag or anti-DNMT1 antibodies by Western blot analysis. (**B**) DNMT1 abundance in the cells derived from various species was measured using Western blot analysis. (**C**) Amino acid homology with selected antigenic peptide sequences of DNMT1 from different species. The red color letters indicate amino acid sites that differ from the antigenic peptide sequence.

**Figure 4 genes-14-01324-f004:**
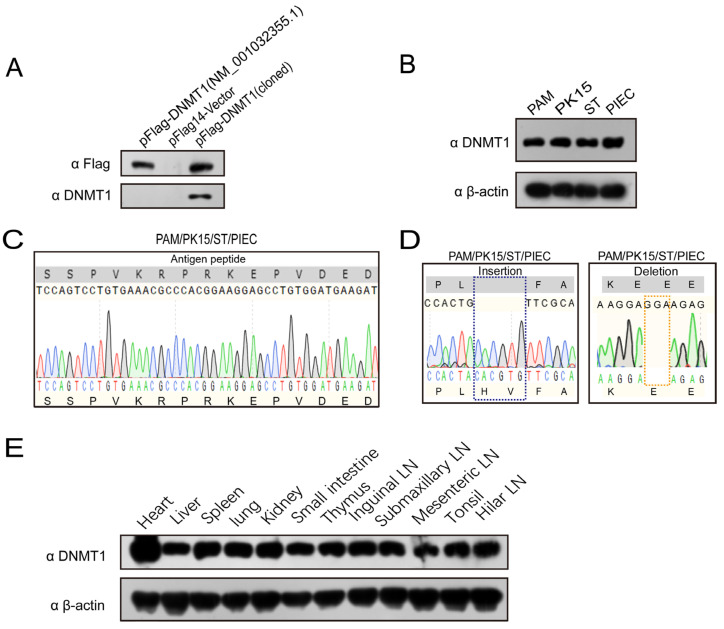
Specific identification of two different porcine DNMT1s. (**A**) BHK-21 cells were transiently transfected with pFlag–DNMT1 (NM_001032355.1), pFlag14–Vector, or pFlag14–DNMT1 (cloned) plasmids. The cell lysates were collected for analysis with an anti-Flag antibody or anti-DNMT1 antibody in a Western blot analysis. (**B**) Lysates of porcine cell lines (PK15, ST, PIEC) and PAMs were harvested to determine their DNMT1 expression with Western blotting. (**C**) Sequencing analysis of various porcine cells based on the selected antigenic peptide sequences. (**D**) Sequencing analysis of the inserted and deleted regions in DNMT1 from various porcine cells compared with that in the previously reported porcine DNMT1 (used as the control). (**E**) Approximately 100 mg of each tissue sample (heart, liver, spleen, lung, kidney, small intestine, thymus, inguinal lymph nodes, submaxillary lymph nodes, mesenteric lymph nodes, tonsils, and hilar lymph nodes) of piglets were obtained. Protein samples were prepared according to the material and method section to determine their DNMT1 expression by Western blotting.

## Data Availability

The data presented in this study are available on request from the corresponding author.

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
