# Peer review of "A Porcine DNMT1 Variant: Molecular Cloning and Generation of Specific Polyclonal Antibody"

_genes, 2023, doi:10.3390/genes14071324_

Round 1

Reviewer 1 Report (Previous Reviewer 1)

The authors have responded to previous review but not in a particularly thoughtful or complete manner.

To what extent does the DNMT1 protein expression detected with the Ab compare to expression in previously published studies using mRNA analysis (RNAseq or RT-PCR or …)?

Again, generating an Ab is not particularly interesting in itself.  Neither is the fact that it does not recognize a DNMT1 peptide that differs by a single amino acid from the antigen used to generate it.  Or that it does not recognize all species tested.

It seems that continuing to use the DNMT1 reference sequence that differs from this cloned sequence (and other database sequences) makes for a complicated presentation.

It may be that the author’s cloned sequence represents the major allele of the pig DNMT1 gene and that the reference sequence is the minor allele.  A most interesting result that could be obtained here would be to genotype many pigs from different breeds to determine if this is true or not.

When referring to the gene, DNMT1 needs to be italicized.

There are spelling and grammatical errors throughout.

Author Response

Comments and Suggestions for Authors

The authors have responded to previous review but not in a particularly thoughtful or complete manner.

Response: We would like to thank you for the thorough and thoughtful review of our manuscript. The concerns raised by the reviewer have been addressed in the revised manuscript or the following response. Additionally, all the minor corrections have been made in the revised manuscript.

To what extent does the DNMT1 protein expression detected with the Ab compare to expression in previously published studies using mRNA analysis (RNAseq or RT-PCR or …)?

Response: We really appreciate the reviewer’s suggestion. For generation of the specific antibody against porcine DNMT1, our goal of this study was to verify the cloning and sequencing results of porcine DNMT1 gene. Of course, we are also very interested in this question and really want to use the polyclonal antibody as a tool to check DNMT1 expression which has been studied by mRNA analysis. Unfortunately, in current stage, we haven’t done any analysis like that yet. In the future, we will use this antibody as a tool for further studies.

Again, generating an Ab is not particularly interesting in itself.  Neither is the fact that it does not recognize a DNMT1 peptide that differs by a single amino acid from the antigen used to generate it.  Or that it does not recognize all species tested.

Response: As mentioned above, the goal of this study is to use the polyclonal antibody as a tool to verify the cloning and sequencing results of porcine DNMT1 genes. In the meanwhile, we have demonstrated characteristics of the polyclonal antibody including its cross-reactivity to DNMT1 from different species. Interestingly, we have demonstrated that a single amino acid difference in the antigenic peptide region of the reported DNMT1 abolished its reactivity to our polyclonal antibody.

It seems that continuing to use the DNMT1 reference sequence that differs from this cloned sequence (and other database sequences) makes for a complicated presentation.

Response: We have revised the description, L166-169 “Since the 8 porcine DNMT1s are staying predicted, we chose the cloned DNMT1, the reported porcine DNMT1 (NP_001027526.1), human DNMT1 (NP_001124295.1), and mouse DNMT1 (NP_001300940.1) for multiple sequence alignment.”

It may be that the author’s cloned sequence represents the major allele of the pig DNMT1 gene and that the reference sequence is the minor allele.  A most interesting result that could be obtained here would be to genotype many pigs from different breeds to determine if this is true or not.

Response: We thank the reviewer’s suggestion. In figure 4 C and 4D, we have determined the antigenic peptide sequence, the insertion region sequence, and the deletion region sequence by PCR using genome DNA from different porcine cells. Those results support the evidence that the cloned DNMT1 should be the only one in those detected cells. In order to understand expression of the cloned DNMT1 and the reported DNMT1 gene, we totally agree that more pigs from different groups are needed to be determined in the future studies.

When referring to the gene, DNMT1 needs to be italicized.

Response: We have checked it up in the revised manuscript.

Comments on the Quality of English Language

There are spelling and grammatical errors throughout.

Response: We have corrected those errors in the revised manuscript.

Reviewer 2 Report (Previous Reviewer 3)

The revised manuscript is ready for publication and my comments have been addressed.

Author Response

Thank you very much for your thoughtful review of our manuscript.

This manuscript is a resubmission of an earlier submission. The following is a list of the peer review reports and author responses from that submission.

Round 1

Reviewer 1 Report

In this report the authors report a sequence of porcine DNMT1 cDNA and the generation of a polyclonal antibody capable of detecting the protein in several porcine cell types.

Simply, they have obtained a cDNA sequence from porcine alveolar macrophages, performed standard sequence comparisons with reference sequences form porcine and other species cDNAs.  They have also generated a polycloncal antibody against the N terminal amino acids that can work across pig immune cells and those of some species, but not the protein expected from the pig database amino acid sequence (which differs by one amino acid).  

Further defining the roles of DNMTs in any species and any tissue would potentially be of interest.  But unfortunately, there is no hypothesis tested using the antibody reagent, so I do feel a limit at this time.  I also consider the sequence analysis and sequence comparisons a bit unclear at this time, particularly surrounding the reference sequence used, where more could be done to clarify the matter of whether this is a “novel” sequence.

Comments.

1.  Section 3.1 and elsewhere.  Terminology.  They have not cloned the gene; they have cloned a cDNA. 

2.  Regarding lines 130 - 132.   “The cloned porcine DNMT1 was 99% identical to the announced porcine DNMT1 (NP_001027526.1) with a 2-amino acid insertion, a 1-amino acid deletion, and 12 amino acid mutations (Figure 1).”

a. There must be porcine DNMT1 cDNA sequences other than NP-001027526.1 available in sequence databases that they can compare their cloned sequence with to determine if their sequence, which differs from the one they are comparing it with, is truly novel?  Also, 12 amino acid substitutions in comparison to the reference sequence is either interesting or troublesome depending on your point of view.  Were these regions re-sequenced to determine if PCR artifacts or other issues are at play?

b.  i.e., scan RNAseq or whole genome sequence databases not just this single reference sequence.

c. Was any 5’ or 3’ UTR sequence obtained to help determine how it relates to other porcine DNMT cDNAs?

d. The possibility of breed differences is brought up (Discussion lines 261 – 271) and really needs to be examined further with sequence data, again, either from existing databases or gathered by the authors themselves. 

3.  Discussion lines 248 – 250. Moreover, this antibody has cross-reactivity with 248 DNMT1 of other species including humans, monkeys, dogs, and cattle. Collectively, these 249 data indicated a novel DNMT1 variant in porcine alveolar macrophages.” The authors should test other cell and tissue types in order to make a statement that this is a novel variant and not just one expressed in porcine alveolar macrophages.

4.  Figure 1 comes through as blurry and very difficult to read.

Author Response

We would like to thank the reviewer for the thorough and thoughtful review of our manuscript. The concerns raised by the reviewer has been addressed pointed-by-pointed.

Reviewer 2 Report

The manuscript by Lin Zhu et al. identified a novel porcine DNMT1 variant. Their data showed that a 2-amino acid insertion, a 1-amino acid deletion, and 12 amino acid mutations in the novel porcine DNMT1. By using polyclonal antibody generated from antigen peptide including these amino acid mutated sites, they distinguish this novel DNMT1 variant from reported porcine DNMT1 in some porcine cells and cells derived from some other species. Generally, the paper is well written and logic in this paper is clear. The discovery of this new DNMT1 variant may broaden the function of DNMT1, which needs to be further clarified. I see no limitations to the study.

Author Response

We would like to thank Reviewer 2 for their kind words and overall positive response to the manuscript. 

Reviewer 3 Report

The study by Zhu et al. discovered a porcine DNMT1 variant which is different from the published one in Genbank. They further produced polyAb against the variant and performed characterizations using a variety of cell lines. The so-called “novel” is claimed by the observation that the pAb recognize endogenous DNMT1 from all tested porcine cells, but not the published “original” DNMT1. Several interesting questions are described as below.

-        What’s the host infor of the published DNMT1? Breed, cell type, etc. Per the broad recognition of the pAb, it’s interesting that all tested porcine cell lines express DNMT1 different than the published one, at least for the epitope binding region.

-        Understand that the peptide was predicted for pAb production. Could you provide the rational of selecting this specific peptide for immunization? Softwares for prediction, criteria for selection, etc? The DNMT1 looks like to be a pretty large protein.

Minor comments:

-        The discussion mentioned the PAM cells are from Pudong white pig. it’s advised to add this infor into methods part to identify the animal source/breed.

-        Fig. 3B shows that pAb does not recognize N2a mouse cell line which might be due to the three AA mutations in the peptide region. Please describe accordingly in the maintext.

Author Response

We would like to thank the reviewer for the thoughtful review of our manuscript. The concerns raised by the reviewer has been addressed point-by-point.

Round 2

Reviewer 1 Report

The authors have pretty much addressed all my comments except one, and that is the most important one.

My understanding now is that the sequence they have generated has been generated many times before so it is not novel.  It's just that the reference sequence differs from this other sequence.  In any event the title should not contain the word novel.

I felt my most important comment has not been addressed and should be seriously considered by the editors.  My original comment follows in quotation marks: "Further defining the roles of DNMTs in any species and any tissue would potentially be of interest.  But unfortunately, there is no hypothesis tested using the antibody reagent, so I do not feel publication of such a limited research product is merited in Genes at this time."

I stand by that original comment.